# Population Distribution and Patients’ Awareness of Food Impaction: A Cross-Sectional Study

**DOI:** 10.3390/healthcare12171688

**Published:** 2024-08-23

**Authors:** Zhe Zhao, Zikang He, Xiang Liu, Qing Wang, Ming Zhou, Fu Wang, Jihua Chen

**Affiliations:** State Key Laboratory of Oral & Maxillofacial Reconstruction and Regeneration, National Clinical Research Center for Oral Diseases, Shaanxi Key Laboratory of Stomatology, Department of Prosthodontics, School of Stomatology, The Fourth Military Medical University, Xi’an 710032, China; 17839927278@163.com (Z.Z.);

**Keywords:** dental diseases, oral health, consultation

## Abstract

Background: Food impaction can contribute to a variety of oral health problems. However, the prevalence of food impaction in the population and patient awareness of these issues are poorly reported on. Methods: A questionnaire about food impaction was designed and uploaded to an online platform (Sojump) which was then circulated among the study participants using various social media platforms. Participants were asked to anonymously respond to the questionnaire regarding the prevalence of food impaction, its influence on their quality of life, their consultation rates and their oral cleaning methods. The survey was conducted through an online survey portal. Statistical analyses were performed using SPSS and GraphPad. The Chi-Square test, Bonferroni test and the Kruskal–Wallis H test were used to measure categorical variables from the survey. Results: The results showed that the prevalence of food impaction in non-dental professional participants was 86.9%. Among these patients, 12,157 pairs/cases of proximal contacts were affected. The number of food impaction cases in posterior teeth was significantly higher than in anterior teeth. Approximately 81.9% of patients believed that food impaction could affect their lives. However, the consultation rate for these patients was only 17.7%. Conclusions: This study revealed that food impaction continued to have a high rate of incidence and a low rate of consultation, potentially due to a lack of awareness regarding its influence on oral health. To effectively prevent and address problems resulting from food impaction, both dentists and society need to enhance oral health knowledge in the population.

## 1. Introduction

Food impaction is a common phenomenon wherein food gets wedged between two teeth due to biting force during chewing [1]. Typically, the integrity of the proximal contact between adjacent teeth serves as a barrier to prevent food from entering the interdental space. This can be achieved through the assistance of marginal ridges, developmental grooves and the contours of the facial and lingual surfaces of the teeth. Once the proximal contact is compromised or unable to withstand the force that separates the surfaces of two adjacent teeth, food impaction may occur.

Patients who experience food impaction often exhibit noticeable subjective symptoms, such as pain and discomfort in the affected area. Food impaction can reduce a patient’s chewing efficiency, increase the burden on their gastrointestinal system and substantially contribute to gingivitis and periodontitis [2,3]. The mechanical irritation exerted on periodontal tissue may cause gingivitis and bone resorption in the impacted area [4]. The impacted food may likely serve as a favorable medium for bacteria accumulation, leading to an exacerbation of damage to both dentition and periodontal tissues [3]. Periodontitis also acts as a risk factor for the development of systemic diseases, including coronary heart disease and diabetes mellitus, when combined with congenital genetic factors and traditional risk factors [5,6]. In addition to natural teeth, food impaction is a commonly encountered issue following restorative or implant therapy [7]. A prospective trial conducted by Lu et al. examined the prognosis of 38 patients who underwent restorative treatment over a period of 1 year. The study revealed that food impaction occurred in 15.62% of cases [8]. The prevalence of food impaction in 779 implant crowns from 489 patients was 16.6% over a period of 3 years. Notably, food impaction occurred more frequently in the premolar and molar areas [9].

To sum up, food impaction is among the important factors that affect oral health. Therefore, it is crucial to clarify the distribution of food impaction among the population and understand the factors that influence its prevalence. In addition, examining how individuals perceive this prevalent oral phenomenon is an important public health concern that deserves attention. However, previous research has lacked comprehensive cross-sectional investigations into this particular topic.

To attain a comprehensive understanding of food impaction, the present study conducted a large sample survey concerning the distribution of food impaction and its impact on quality of life, consultation rates, the curative effectiveness of treatments and oral hygiene practices related to food impaction.

## 2. Materials and Methods

A questionnaire was designed based on relevant literatures and expert suggestions, which consisted of four parts: demographic data, population prevalence, patient awareness and oral cleaning methods related to food impaction. The first part included information such as age, gender, profession and other personal details. The second part included factors influencing the prevalence of food impaction. The third part included the degree to which food impaction affects quality of life, consultation rates and patient evaluations of the curative effectiveness of treatments. The fourth part included the methods and efficiency of cleaning impacted food. With the exception of the question concerning the participant’s age, which required a fill-in-the-blank response, all other questions were presented as multiple-choice. Each participant took approximately 5–10 min to complete the questionnaire.

The survey was conducted anonymously using an online survey portal. The study was conducted according to the guidelines of the Declaration of Helsinki and was approved by the Ethical Committee of the Stomatological Hospital of the Fourth Military Medical University (approval no. IRB-REV-2021047). Informed consent was obtained from all subjects involved in the study. The survey included adult participants ranging in age from 18 to 78 years old.

All statistical analyses were performed using SPSS v23 (SPSS Inc., Chicago, IL, USA) and GraphPad v8.0.2 (GraphPad Prism Software Inc., San Diego, CA, USA). Chi-Square, Bonferroni and Kruskal–Wallis H tests were used to measure categorical variables from the survey. The predetermined level of statistical significance for all analyses was set at α = 0.05.

## 3. Results

### 3.1. Demographic Characteristics

Among all the study participants, some individuals were students or employees affiliated with dental schools or hospitals (classified as the dental profession group), while the remaining individuals were not associated with these institutions (classified as the non-dental profession group). Notably, the dental profession group had a prevalence rate of 76.6%, while the non-dental profession group had a prevalence rate of 86.9%. A significant difference was observed in the prevalence rates of food impaction between the two groups (*p* < 0.001). The dental profession group had a consultation rate of 26.2%, while the non-dental profession group had a consultation rate of 15.4%. These results revealed that the consultation rate of the dental profession group was significantly higher than that of the non-dental profession group (*p* < 0.001). In addition, there were also significant differences in cleaning methods between the dental profession and non-dental profession groups (*p* < 0.001). Among dental professionals, dental floss was the most frequently used method, while among non-dental professionals, toothpicks were the most frequently used method. To avoid bias due to the participation of dental professionals, we chose non-dental professional participants as the population for the following analysis.

The demographic characteristics of non-dental professional participants are summarized in Table 1. A total of 3581 non-dental professional participants were investigated, including 1486 males (41.5%) and 2095 females (58.5%).

### 3.2. Influencing Factors on the Prevalence of Food Impaction

Out of the 3581 participants surveyed, 3111 suffered from food impaction. The overall prevalence of food impaction was 86.9%. There was a statistical difference in the prevalence of food impaction among different age groups (*p* < 0.001) (Figure 1a). The prevalence rate of food impaction increased with age until the 58–67 age group, after which it slightly decreased in the 68–78 age group. The Bonferroni test showed that the prevalence of food impaction was lowest in the 18–27 age group, which was statistically different from other age groups (*p* < 0.05). A significant difference was observed in the prevalence rates of food impaction between genders (*p* < 0.05). However, after controlling for the age factor, the results showed that there was no statistically significant difference in the prevalence of food impaction between males and females (*p* > 0.05) (Figure 1b).

Among the 3111 patients identified with food impaction, a total of 12,157 cases of proximal contacts were affected, with an average of 4 cases of proximal contacts per person. The ratio of food impaction cases in posterior teeth was higher than in anterior teeth (Figure 2a). The distribution of food impaction in patients of different genders were analyzed using the Bonferroni test. It was found that the percentage of food impaction in the upper anterior and left upper posterior teeth was slightly higher in males than in females (*p* < 0.05). Conversely, the percentage of food impaction in the right lower posterior tooth was slightly higher in females than in males (*p* < 0.05). No significant differences were found in other tooth positions (*p* > 0.05) between the two groups (Figure 2b).

The component ratio of posterior teeth was higher than that of anterior teeth. As age increased, the impaction ratio of anterior teeth decreased, while the impaction ratio of posterior teeth increased (Figure 2c). Proximal contact between the first and second molars was the most frequently impacted area, and the proportion of proximal contact between the two central incisors was the highest for the anterior region (Figure 2d).

### 3.3. Patients’ Awareness and Coping Attitudes towards Food Impaction

The impact of food impaction on the quality of life of participants was classified into five grades: Level 1—basically no impact, Level 2—slight impact, Level 3—impacted but tolerable, Level 4—unbearable and Level 5—needs immediate attention. According to the results presented in Table 2, it was found that Level 1 accounted for 1.9% of participants, Level 2 accounted for 16.2%, Level 3 accounted for 48.5%, Level 4 accounted for 13.5% and Level 5 accounted for 19.8%. Levels 1 and 2 were classified as having no significant impact on life, while Levels 3–5 were classified as having a significant impact on participants’ lives.

As the number of food impacted proximal contacts increased, there was a corresponding increase in the component ratio of Levels 3–5, indicating a higher degree of impact on individuals’ lives (Figure 3a). This suggests that the higher the number of proximal contacts affected by food impaction, the more severe the impact will be on an individuals’ overall quality of life. In examining the influence of tooth position on the impact of food impaction on patients’ lives, we specifically focused on individuals who had only one impacted proximal contact. By doing so, we aimed to exclude the potential influencing factor of the number of proximal contacts. The results of the study showed that the proportion of individuals classified under Levels 3–5 was higher in the posterior region compared to the anterior region (Figure 3b). Therefore, based on these findings, it can be concluded that the impact of food impaction in the posterior teeth area has a more severe impact on individuals’ lives and is deemed more unbearable. The results of the Kruskal–Wallis H Test showed that posterior teeth had a higher mean level of impact on life than anterior teeth (*p* < 0.001). However, it is important to note that among those whose lives were significantly affected by food impaction, the actual consultation rate was only 17.7%.

A total of 479 patients received dental treatment for food impaction in this survey. An evaluation of the curative effect for these patients revealed that only 20.3% of them considered the treatment to be very effective (Figure 4a). There was no significant difference in the curative effect of food impaction treatment among different tooth positions (*p* > 0.05) (Figure 4b).

### 3.4. Oral Cleaning Methods for Food Impaction Patients

Regarding the cleaning methods of food impaction patients, the highest proportion (32.2%) reported using toothpicks. This was followed by dental floss (31.8%), brushing and rinsing (28.3%), tooth punch (7.4%) and a small percentage (0.3%) who reported not using any treatment. According to the statistics based on age groups, there were differences in preference for cleaning methods in different age groups (*p* < 0.001) (Figure 5a). People aged 18–47 had the highest proportion of using dental floss for cleaning, while individuals over the age of 48 were more inclined to choose toothpicks as a cleaning method. There was a significant difference between the cleaning methods chosen by men and women (*p* < 0.001) (Figure 5b). Men had a higher usage of toothpick compared to women, while women had a higher usage of floss and tooth punch compared to men. When comparing the effectiveness of oral cleaning methods for patients with food impaction, we specifically focused on individuals who used only one kind of cleaning method. By doing so, we aimed to exclude the potential influencing factor of the number of cleaning methods. The results show that dental floss and tooth punch were found to be more effective than brushing and rinsing, as well as using toothpicks (Figure 5c).

## 4. Discussion

Oral disease has emerged as a significant public health issue globally [10]. It has been reported that 3.9 billion people worldwide are affected by oral diseases [11]. The interrelationship between oral disease and overall health emphasizes the significance of oral health as an integral part of overall wellbeing [12]. In today’s world, inequality in oral healthcare and maintenance continues to grow, resulting in a persistently high prevalence of oral diseases and a significant socioeconomic burden associated with them [13]. Food impaction is indeed a common oral disease. In the surveyed population, the prevalence rate of food impaction in non-dental professional participants was found to be 86.9%. This high prevalence was observed across different age groups. Food impaction has often been overlooked in comparison to dental caries and periodontitis, primarily due to its insidious symptoms. Nevertheless, the widespread occurrence of food impaction in the population, coupled with its mutually reinforcing relationship with the development and progression of dental caries and periodontal disease, highlights its potential risk to oral health.

The increased prevalence of food impaction with increasing age may indeed be related to changes in the proximal contact and occlusal surface of teeth as individuals age [14,15,16]. Natural teeth undergo a physiological movement of 56–108 μm during chewing [17]. This movement causes the adjacent proximal surfaces of teeth to wear against each other, resulting in changes to the shape of the proximal contact [18]. In response to this wear, teeth tend to experience a mesial drift, which helps compensate for reduction in proximal contact [19]. Once any factor disrupts this balance, it can result in a reduction of the tightness of the proximal contact between adjacent teeth, leading to the occurrence of food impaction. In addition, the natural attrition of the occlusal surface with age is also an important factor that can affect the occurrence of food impaction. Tooth attrition can lead to changes in the occlusal contact center and occlusal spillway. When the occlusal center is located in the proximal contact area or the occlusal spillway is lost, it can create conditions where food becomes wedged between two teeth. This can occur due to abnormal occlusal force or a failure to spill properly during chewing. The prevalence of food impaction tends to decrease in individuals aged 68–78, which may be attributed to tooth loss that commonly occurs in older adults. According to data from the Centers for Disease Control and Prevention, approximately one-sixth of older adults aged over 65 in the United States have lost all of their teeth [20]. In China, according to the results of the Fourth National Oral Health Survey conducted from 2013 to 2015, there has been a rising trend in the prevalence of edentulism among the older adult population [10]. With the loss of teeth, the presence of proximal contact gradually disappears. When an individual has no remaining teeth in their mouth, the condition of proximal contact no longer exists. Consequently, the prevalence of food impaction is expected to decrease in such cases.

The effect of the dental profession on the prevalence of food impaction is likely to be associated with the acquisition of oral health knowledge and oral health literacy. Oral health knowledge is widely recognized as a prerequisite for maintaining proper oral healthcare-related behaviors [21,22,23]. Oral health literacy (OHL) is defined as the extent to which an individual has the capacity to obtain, process and comprehend basic oral health information and services in order to make appropriate decisions regarding oral healthcare [24,25]. For individuals engaged in the field of dentistry, it is expected that they would acquire a higher level of oral health knowledge and, consequently, have a higher level of OHL.

During the process of chewing, the anterior teeth play a primary role in tearing food, while the molars are responsible for grinding food. The chewing center is predominantly located in the posterior teeth [26]. Moreover, food impaction is commonly found between the first and second molars. In the conducted survey, when analyzing the location of food impaction, the most frequent occurrence was also found in the proximal contact area between the first and second molars (Figure 2d). It is speculated that the reason may be attributed to several factors. Firstly, the first molars serve as the chewing centers of the dentition. Secondly, the second molars often appear as terminal teeth. The distal displacement of the second molar can be caused by various abnormal occlusal factors, contributing to the occurrence of food impaction [8].

Food impaction can lead to a series of oral health problems that can have varying degrees of impact on a patient’s life. In this survey, the impact of food impaction on life was categorized into five levels. The highest proportion of respondents reported that food impaction had an impact on their life, but that it was tolerable. This finding is related to the clinical features of food impaction. In the population with mild symptoms, the clinical manifestations of food impaction included gingival pain, redness and swelling in the food impaction area [27]. Only when the frequency of food impaction increases, various problems such as gingival retraction, loose teeth and dental caries may occur. Therefore, patients tend to exhibit a high degree of tolerance when the symptoms of food impaction do not cause damage to dental or periodontal tissues. This aspect is also reflected in the consultation rate results from the survey. Among the 3111 non-dental professional food impaction patients surveyed, only 479 individuals went to the hospital for food impaction treatment, accounting for 15.4% of the patients. Even for those patients whose lives had already been significantly impacted by food impaction, the consultation rate was only 17.7%. In this study, we also found that the consultation rate among individuals who believed that food impaction had a significant impact on their lives was low. In contrast, patients in the dental profession group had a significantly higher consultation rate. This may be related to the lack of OHL in the study population. Previous research has shown that low levels of OHL are associated with poor dental behaviors [24]. Findings from a survey conducted among pharmacy students showed that despite being medical students, most of the pharmacy students only went to see a dentist when they experienced significant discomfort [28]. Individuals in the dental profession group possessed higher OHL levels compared to those in the non-dental profession group, and therefore their consultation rate was significantly higher (*p* < 0.0001). To promote the consultation rate of patients with food impaction, it is necessary to improve the OHL levels in the general population.

Among the 479 patients who received treatment at the clinic, only 20.3% of them perceived the treatment to be highly effective. The majority of patients were not fully satisfied with the effect of their treatment. It is the most fundamental requirement to develop a treatment plan based on the identified causes [29]. For food impactions with clearly identifiable causes, such as tooth defects, loss of proximal contact, gingival recession, etc., appropriate treatment measures can be implemented to solve these problems. These measures can include fillings, restorations, orthodontic treatment or artificial gingival procedures [29,30,31]. However, there is no clear identifiable cause for a large portion of food impaction cases. For individuals in whom the cause of food impaction is unclear, there may be no basis for clinical treatment. And the absence of a clear basis for clinical treatment may be the reason why many patients express dissatisfaction with the curative effect of their treatment.

Given the uncertainty surrounding the current effectiveness of treatments, the ultilization of oral cleaning tools to promptly remove impacted food is not only indispensable, but is also a beneficial compensation. However, a considerable number of people in China still lack an awareness of effective methods for cleaning the proximal contacts of teeth, and are unfamiliar with the correct usage of dental floss. These results demonstrate that flossing and punching are more effective methods to clean food impaction. However, many patients still do not choose these optimal methods. The survey comparing oral cleaning methods among dental and non-dental professionals revealed a higher rate of flossing ultilization among dental professionals. Therefore, it can be inferred that training experience in stomatology schools or hospitals may influence the selection of these oral maintenance methods. It has been reported that dentists and healthcare professionals are the primary sources of oral health information for patients [32]. Hence, it is necessary for dentists to educate patients with food impaction on oral cleaning methods.

## 5. Conclusions

Food impaction has an extremely high prevalence within the population, and it adversely affects the lives of patients. However, the consultation rate for food impaction is extremely low. At the same time, due to the complex etiology and the varieties of food impaction, the therapeutic effect varies greatly across different types. It is necessary for dentists to study the causes of various types of food impaction and expeditiously put forward effective treatments tailored to address these causes. At the same time, it is also essential for dentists to improve oral health education among patients and within the community. When necessary, it is crucial to carry out interprofessional education and collaborative practice [33], involving nurses, primary healthcare workers and other allied health professionals to educate and disseminate knowledge widely to a broader population. This study used an online questionnaire and therefore the contents of the survey were limited. In order to achieve a more in-depth study of food impaction, clinical examination and analysis of food impacted patients are also needed. This is the next phase of the experiment and we are working on it.

## Figures and Tables

**Figure 1 healthcare-12-01688-f001:**
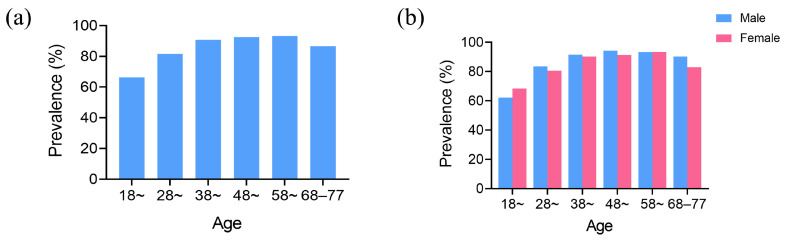
Analysis of the prevalence of food impaction. (**a**) The prevalence of food impaction in different age groups. (**b**) The prevalence of food impaction among males and females in different age groups.

**Figure 2 healthcare-12-01688-f002:**
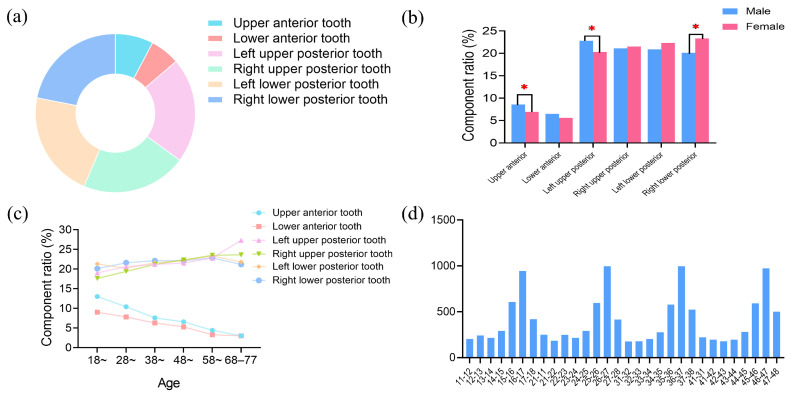
Analysis of the tooth positions and proximal contact of food impaction. (**a**) Distribution of food impaction in different tooth positions. (**b**) The tooth positions in male and female patients. (**c**) The tooth positions in patients of different ages. (**d**) The number of food impaction cases in each proximal contact. The asterisk denotes statistically significant differences * *p* < 0.05.

**Figure 3 healthcare-12-01688-f003:**
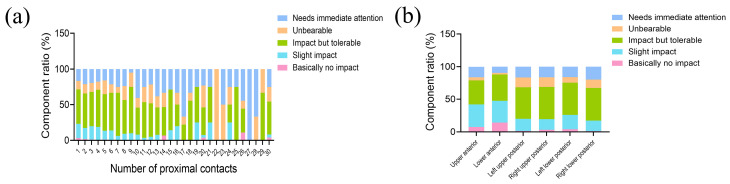
The impact of food impaction on quality of life. (**a**) Analysis of the impact of food impaction on quality of life based on the number of proximal contacts. (**b**) Analysis of the impact of food impaction on quality of life based on different tooth positions.

**Figure 4 healthcare-12-01688-f004:**
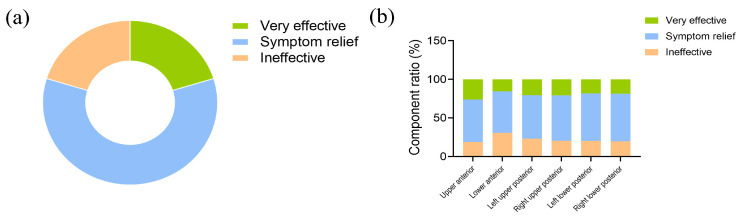
Consultation rate and curative effect of food impaction. (**a**) Evaluation of curative effect in patients with food impaction. (**b**) Analysis of the curative effect of food impaction treatment among different tooth positions.

**Figure 5 healthcare-12-01688-f005:**
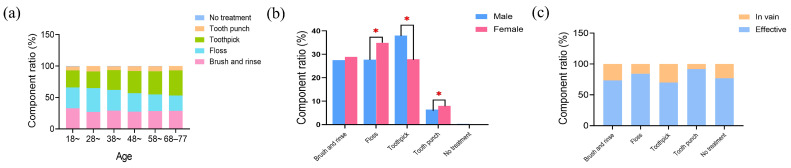
Oral cleaning methods for food impaction patients. (**a**) Analysis of oral cleaning methods for patients of different ages. (**b**) Analysis of oral cleaning methods for patients of different genders. (**c**) Evaluation of the effectiveness of oral cleaning methods in patients with food impaction. The asterisk denotes statistically significant differences * *p* < 0.05.

**Table 1 healthcare-12-01688-t001:** The number and proportion of each group of the population.

Variable	Category	N (%)
Age	18–27	375 (10.5%)
	28–37	790 (22.1%)
	38–47	997 (27.8%)
	48–57	918 (25.6%)
	58–67	419 (11.7%)
	68–78	82 (2.3%)
Gender	male	1486 (41.5%)
	female	2095 (58.5%)

**Table 2 healthcare-12-01688-t002:** Evaluation of the impact of food impaction on patients’ lives.

Evaluation of the Impact of Life	N	Component Ratio (%)
Basically no impact	60	1.9
Slight impact	504	16.2
Impacted but tolerable	1501	48.5
Unbearable	421	13.5
Needs immediate attention	616	19.8
Total	3111	100.0

## Data Availability

The datasets used and/or analyzed in this study are available from the corresponding author on reasonable request.

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
