# Peer review of "Population Distribution and Patients’ Awareness of Food Impaction: A Cross-Sectional Study"

_healthcare, 2024, doi:10.3390/healthcare12171688_

Round 1

Reviewer 1 Report

Comments and Suggestions for Authors

The weaknesses of this study should be emphasized more. Although there was no methodological error, the addition of clinical parameters would have made the study stronger.

This manuscript is of a nature that will attract the reader's attention in terms of its subject matter.

This manuscript does not have much unclear text, but the following minor corrections are necessary.

Abstract: The method section should be expanded.

Materials and methods: It is nice that it was studied in 6903 patients. However, it would have been meaningful to have other clinical parameters and a larger sample size (DMFT, ICDAS, etc.)

Discussion: Although this section is well written, there are some shortcomings. For example, the limitations of the study are not mentioned at all.

References: References 1, 6, 16, and 31 are very old. They should be replaced with new ones.

Comments on the Quality of English Language

There is no major problem with the language. The text is understandable.

Author Response

We want to thank reviewer for constructive and insightful criticism and suggestion. We addressed all the points raised by the reviewer as summarized below.

Question 1: Abstract: The method section should be expanded.

Our response: We have added a description of the method section as followed “A questionnaire based on food impaction was prepared and uploaded in an online platform (Sojump) which was circulated among the participants using various social media platforms. Participants were asked to anonymously respond to a questionnaire regarding the prevalence of food impaction, its influence on quality of life, consultation rates and oral cleaning methods. The survey was conducted through an online survey portal. Statistical analyses were performed using SPSS and GraphPad. The Chi-Square test, Bonferroni test and the Kruskal-Wallis H test were used to measure categorical variables from the survey.”

Question 2: Materials and methods: It is nice that it was studied in 6903 patients. However, it would have been meaningful to have other clinical parameters and a larger sample size (DMFT, ICDAS, etc.)

Our response: It's a very kind suggestion that we will carry out this experiment in our next study. And this work is in progress.

Question 3: Discussion: Although this section is well written, there are some shortcomings. For example, the limitations of the study are not mentioned at all.

Our response: We have added a description of the limitations of the study to the article as follows, "This study used an online questionnaire and therefore the contents of the survey were limited. In order to achieve a more in-depth study of food impaction, clinical examination and analysis of food impacted patients are also needed. This is the next phase of the experiment and we are working on it.”

Question 4: References: References 1, 6, 16, and 31 are very old. They should be replaced with new ones.

Our response: Thanks for your suggestion and we have replaced these references.

Reviewer 2 Report

Comments and Suggestions for Authors

RE: Population Distribution and Patient Awareness of Food Impaction: A Cross-Sectional Study

The reviewer's comments are as follows;

1. Please describe the method (online survey?) for collecting data in the abstract.

2. The online survey was done for all Chinese population or for population in restricted areas?

3. When was dis study conducted?

4. Is there no limitation in the study?

5. Almost half (48.1%) of the respondents were stomatology profession.  This is not a representative of the general population.

Author Response

Thank you very much for your careful review and constructive suggestions with regard to our manuscript. I have made comprehensive and detailed changes according to your suggestions. 

Question 1: Please describe the method (online survey?) for collecting data in the abstract.
Our response: Yes, this experiment was conducted through an online survey. We have added a description of the method section of the abstract as followed “A questionnaire based on food impaction was prepared and uploaded in an online platform (Sojump) which was circulated among the participants using various social media platforms. Participants were asked to anonymously respond to a questionnaire regarding the prevalence of food impaction, its influence on quality of life, consultation rates and oral cleaning methods. The survey was conducted through an online survey portal. ”

Question 2: The online survey was done for all Chinese population or for population in restricted areas?

Our response: The online survey population is not restricted by area.

Question 3: When was dis study conducted?

Our response: This survey was carried out from January 2022 to March 2022.

Question 4: Is there no limitation in the study?

Our response: We have added a description of the limitations of the study to the article as follows, "This study used an online questionnaire and therefore the contents of the survey were limited. In order to achieve a more in-depth study of food impaction, clinical examination and analysis of food impacted patients are also needed. This is the next phase of the experiment and we are working on it.”

Question 5: Almost half (48.1%) of the respondents were stomatology profession. This is not a representative of the general population.

Our response: Thank you for the comments and we have made extensive modifications to our manuscript. To avoid bias due to the participation of dental professionals, we chose non-dental professional participants as the population to be analysed.  

Reviewer 3 Report

Comments and Suggestions for Authors

Although I congratulate to you for your work, this research contains major methodological errors. Although it is a study with a very high sample size, there are fundamental methodological errors. First of all, not homogeneous distribution of the population (a group of dental individuals and a group of non-dental individuals) creates bias and error in the results. While non-dental individuals do not have knowledge about proximal surfaces, dental individuals certainly have knowledge about food impaction. On the other hand, non-dental individuals cannot distinguish food impaction. Additionally, food impaction cannot be diagnosed through a survey alone. Pain caused by an proximal caries or pulpitis is most often confused with food impaction pain. In addition, the distinction between periodonal and dental origins is very important in this regard. Evaluating all these differences through a survey is a fundamental and major mistake. Moreover, as I mentioned at the beginning, dental individuals can distinguish this, while other individuals do not have information about it. This causes heterogeneity of the population and therefore bias. As a result, the accuracy of the results is questionable. In addition, in such a large sample group, advanced statistical methods should be used rather than simple statistical analyses.

Author Response

Thank you very much for your careful review and constructive suggestions with regard to our manuscript. Meanwhile, I am also very grateful to you for the specific modification strategy.  I have made comprehensive and detailed changes according to your suggestions.

Question : Although I congratulate to you for your work, this research contains major methodological errors. Although it is a study with a very high sample size, there are fundamental methodological errors. First of all, not homogeneous distribution of the population (a group of dental individuals and a group of non-dental individuals) creates bias and error in the results. While non-dental individuals do not have knowledge about proximal surfaces, dental individuals certainly have knowledge about food impaction. On the other hand, non-dental individuals cannot distinguish food impaction. Additionally, food impaction cannot be diagnosed through a survey alone. Pain caused by an proximal caries or pulpitis is most often confused with food impaction pain.In addition, the distinction between periodonal and dental origins is very important in this regard. Evaluating all these differences through a survey is a fundamental and major mistake. Moreover, as I mentioned at the beginning, dental individuals can distinguish this, while other individuals do not have information about it. This causes heterogeneity of the population and therefore bias. As a result, the accuracy of the results is questionable. In addition, in such a large sample group, advanced statistical methods should be used rather than simple statistical analyses. 

Our response: Thank you for your kind suggestion and we have made extensive modifications to our manuscript. To avoid bias due to the professional factor, we chose non-dental professional participants as the population to be analysed.

And in addition to pain, food fibres implanted between two teeth is also an important symptom. Because food impaction is a disease that occurs in the process of chewing, it is difficult to maintain the state of food impaction in the clinic. Therefore, most of the patients who come to the clinic with the complaint of food impaction do not show the actual symptoms of impaction in clinical examination. And that is the reason why we chose to do a cursory analysis of food impaction through an online survey.

To address the possible problem of bias due to participants' lack of knowledge about food impaction. Firstly, we provided a detailed description of food impaction on the first page of the questionnaire. In addition, in order to prevent patients from choosing options randomly due to poor knowledge of food impaction or dentistry, we designed a "skip this question" function. Those who did not complete all the questions in the questionnaire were considered invalid and were not included in the statistical analyses. 

Besides, we have added a description of the limitations of the study to the article as follows, "This study used an online questionnaire and therefore the contents of the survey were limited. In order to achieve a more in-depth study of food impaction, clinical examination and analysis of food impacted patients are also needed. This is the next phase of the experiment and we are working on it.”

Round 2

Reviewer 2 Report

Comments and Suggestions for Authors

None.

Author Response

Thanks for the kind and insightful comments and suggestions! 

Reviewer 3 Report

Comments and Suggestions for Authors

Not applicable.

Comments on the Quality of English Language

Not applicable. 

Author Response

We sincerely thank you for the insightful and constructive comments, which will all prove invaluable in the revision and improvement of our paper, as well as in guiding our research in the future. According to the comments, We carefully examined the full manuscript, adjusted the structure of the article, described the methods in more detail, and checked and optimized the data and descriptions in the results section. We also added a description of the limitations of the study to the article and the future plans to improve it in the conclusion. We appreciate for your warm work earnestly and hope that the correction will meet with approval.